# Total Variation Classes Beyond 1d: Minimax Rates, and the Limitations of Linear Smoothers

**Veeranjaneyulu Sadhanala**
Machine Learning Department
Carnegie Mellon University
Pittsburgh, PA 15213
vsadhana@cs.cmu.edu

**Yu-Xiang Wang**
Machine Learning Department
Carnegie Mellon University
Pittsburgh, PA 15213
yuxiangw@cs.cmu.edu

**Ryan J. Tibshirani**
Department of Statistics
Carnegie Mellon University
Pittsburgh, PA 15213
ryantibs@stat.cmu.edu

## Abstract

We consider the problem of estimating a function defined over $n$ locations on a $d$-dimensional grid (having all side lengths equal to $n^{1/d}$). When the function is constrained to have discrete total variation bounded by $C_n$, we derive the minimax optimal (squared) $\ell_2$ estimation error rate, parametrized by $n, C_n$. Total variation denoising, also known as the fused lasso, is seen to be rate optimal. Several simpler estimators exist, such as Laplacian smoothing and Laplacian eigenmaps. A natural question is: can these simpler estimators perform just as well? We prove that these estimators, and more broadly all estimators given by linear transformations of the input data, are suboptimal over the class of functions with bounded variation. This extends fundamental findings of Donoho and Johnstone [12] on 1-dimensional total variation spaces to higher dimensions. The implication is that the computationally simpler methods cannot be used for such sophisticated denoising tasks, without sacrificing statistical accuracy. We also derive minimax rates for discrete Sobolev spaces over $d$-dimensional grids, which are, in some sense, smaller than the total variation function spaces. Indeed, these are small enough spaces that linear estimators can be optimal—and a few well-known ones are, such as Laplacian smoothing and Laplacian eigenmaps, as we show. Lastly, we investigate the adaptivity of the total variation denoiser to these smaller Sobolev function spaces.

## 1 Introduction

Let $G = (V, E)$ be a $d$-dimensional grid graph, i.e., lattice graph, with equal side lengths. Label the nodes as $V = \{1, \ldots, n\}$, and edges as $E = \{e_1, \ldots, e_m\}$. Consider data $y = (y_1, \ldots, y_n) \in \mathbb{R}^n$ observed over the nodes, from a model

$$y_i \sim N(\theta_{0,i}, \sigma^2), \quad \text{i.i.d., for } i = 1, \ldots, n, \tag{1}$$

where $\theta_0 = (\theta_{0,1}, \ldots, \theta_{0,n}) \in \mathbb{R}^n$ is an unknown mean parameter to be estimated, and $\sigma^2 > 0$ is the marginal noise variance. It is assumed that $\theta_0$ displays some kind of regularity over the grid $G$, e.g., $\theta_0 \in \mathcal{T}_d(C_n)$ for some $C_n > 0$, where

$$\mathcal{T}_d(C_n) = \big\{ \theta : \|D\theta\|_1 \le C_n \big\}, \tag{2}$$

and $D \in \mathbb{R}^{m \times n}$ is the edge incidence matrix of $G$. This has $\ell$th row $D_\ell = (0, \ldots, -1, \ldots, 1, \ldots, 0)$, with a $-1$ in the $i$th location, and $1$ in the $j$th location, provided that the $\ell$th edge is $e_\ell = (i, j)$ with $i < j$. Equivalently, $L = D^T D$ is the graph Laplacian matrix of $G$, and thus

$$\|D\theta\|_1 = \sum_{(i,j) \in E} |\theta_i - \theta_j|, \quad \text{and} \quad \|D\theta\|_2^2 = \theta^T L \theta = \sum_{(i,j) \in E} (\theta_i - \theta_j)^2.$$

We will refer to the class in (2) as a *discrete total variation (TV) class*, and to the quantity $\|D\theta_0\|_1$ as the discrete total variation of $\theta_0$, though for simplicity we will often drop the word "discrete".

The problem of estimating $\theta_0$ given a total variation bound as in (2) is of great importance in both nonparametric statistics and signal processing, and has many applications, e.g., changepoint detection for 1d grids, and image denoising for 2d and 3d grids. There has been much methodological and computational work devoted to this problem, resulting in practically efficient estimators in dimensions 1, 2, 3, and beyond. However, theoretical performance, and in particularly optimality, is only really well-understood in the 1-dimensional setting. This paper seeks to change that, and offers theory in $d$-dimensions that parallel more classical results known in the 1-dimensional case.

**Estimators under consideration.**   Central role to our work is the *total variation (TV) denoising* or *fused lasso* estimator (e.g., [21, 25, 7, 15, 27, 23, 2]), defined by the convex optimization problem

$$\hat{\theta}^{\text{TV}} = \underset{\theta \in \mathbb{R}^n}{\operatorname{argmin}} \ \|y - \theta\|_2^2 + \lambda \|D\theta\|_1, \tag{3}$$

where $\lambda \geq 0$ is a tuning parameter. Another pair of methods that we study carefully are *Laplacian smoothing* and *Laplacian eigenmaps*, which are most commonly seen in the context of clustering, dimensionality reduction, and semi-supervised learning, but are also useful tools for estimation in a regression setting like ours (e.g., [3, 4, 24, 30, 5, 22]). The Laplacian smoothing estimator is given by

$$\hat{\theta}^{\text{LS}} = \underset{\theta \in \mathbb{R}^n}{\operatorname{argmin}} \ \|y - \theta\|_2^2 + \lambda \|D\theta\|_2^2, \quad \text{i.e.,} \quad \hat{\theta}^{\text{LS}} = (I + \lambda L)^{-1} y, \tag{4}$$

for a tuning parameter $\lambda \geq 0$, where in the second expression we have written $\hat{\theta}^{\text{LS}}$ in closed-form (this is possible since it is the minimizer of a convex quadratic). For Laplacian eigenmaps, we must introduce the eigendecomposition of the graph Laplacian, $L = V \Sigma V^T$, where $\Sigma = \operatorname{diag}(\rho_1, \ldots, \rho_n)$ with $0 = \rho_1 < \rho_2 \leq \ldots \leq \rho_n$, and where $V = [V_1, V_2, \ldots, V_n] \in \mathbb{R}^{n \times n}$ has orthonormal columns. The Laplacian eigenmaps estimator is

$$\hat{\theta}^{\text{LE}} = V_{[k]} V_{[k]}^T y, \quad \text{where} \quad V_{[k]} = [V_1, V_2, \ldots, V_k] \in \mathbb{R}^{n \times k}, \tag{5}$$

where now $k \in \{1, \ldots, n\}$ acts as a tuning parameter.

Laplacian smoothing and Laplacian eigenmaps are appealing because they are (relatively) simple: they are just linear transformations of the data $y$. Indeed, as we are considering $G$ to be a grid, both estimators in (4), (5) can be computed very quickly, in nearly $O(n)$ time, since the columns of $V$ here are discrete cosine transform (DCT) basis vectors when $d = 1$, or Kronecker products thereof, when $d \geq 2$ (e.g., [9, 17, 20, 28]). The TV denoising estimator in (3), on the other hand, cannot be expressed in closed-form, and is much more difficult to compute, especially when $d \geq 2$, though several advances have been made over the years (see the references above, and in particular [2] for an efficient operator-splitting algorithm and nice literature survey). Importantly, these computational difficulties are often worth it: TV denoising often practically outperforms $\ell_2$-regularized estimators like Laplacian smoothing (and also Laplacian eigenmaps) in image denoising tasks, as it is able to better preserve sharp edges and object boundaries (this is now widely accepted, early references are, e.g., [1, 10, 8]). See Figure 1 for an example, using the often-studied "cameraman" image.

In the 1d setting, classical theory from nonparametric statistics draws a clear distinction between the performance of TV denoising and estimators like Laplacian smoothing and Laplacian eigenmaps. Perhaps surprisingly, this theory has not yet been fully developed in dimensions $d \geq 2$. Arguably, the comparison between TV denoising and Laplacian smoothing and Laplacian eigenmaps is even more interesting in higher dimensions, because the computational gap between the methods is even larger (the former method being much more expensive, say in 2d and 3d, than the latter two). Shortly, we review the 1d theory, and what is known in $d$-dimensions, for $d \geq 2$. First, we introduce notation.

**Notation.**   For deterministic (nonrandom) sequences $a_n, b_n$ we write $a_n = O(b_n)$ to denote that $a_n/b_n$ is upper bounded for all $n$ large enough, and $a_n \asymp b_n$ to denote that both $a_n = O(b_n)$ and $a_n^{-1} = O(b_n^{-1})$. Also, for random sequences $A_n, B_n$, we write $A_n = O_{\mathbb{P}}(B_n)$ to denote that $A_n/B_n$ is bounded in probability. We abbreviate $a \wedge b = \min\{a, b\}$ and $a \vee b = \max\{a, b\}$. For an estimator $\hat{\theta}$ of the parameter $\theta_0$ in (1), we define its mean squared error (MSE) to be

$$\text{MSE}(\hat{\theta}, \theta_0) = \frac{1}{n} \|\hat{\theta} - \theta_0\|_2^2.$$

| Noisy image | Laplacian smoothing | TV denoising |

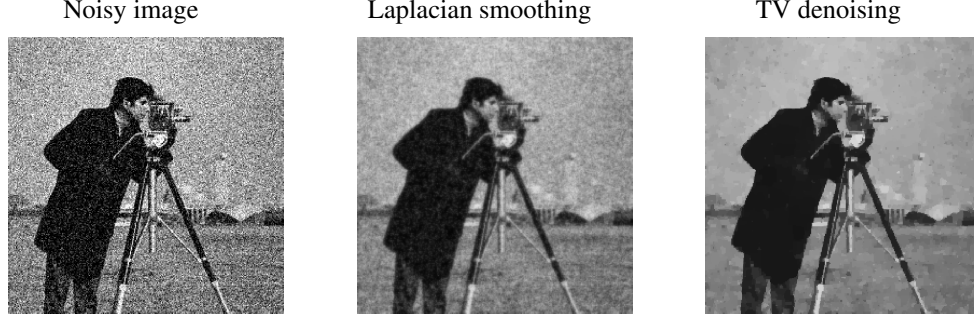

Figure 1: *Comparison of Laplacian smoothing and TV denoising for the common "cameraman" image. TV denoising provides a more visually appealing result, and also achieves aboutx a 35% reduction in MSE compared to Laplacian smoothing (MSE being measured to the original image). Both methods were tuned optimally.*

The risk of $\hat{\theta}$ is the expectation of its MSE, and for a set $\mathcal{K} \subseteq \mathbb{R}^n$, we define the minimax risk and minimax linear risk to be

$$R(\mathcal{K}) = \inf_{\hat{\theta}} \sup_{\theta_0 \in \mathcal{K}} \mathbb{E}\big[\mathrm{MSE}(\hat{\theta}, \theta_0)\big] \quad \text{and} \quad R_L(\mathcal{K}) = \inf_{\hat{\theta} \text{ linear}} \sup_{\theta_0 \in \mathcal{K}} \mathbb{E}\big[\mathrm{MSE}(\hat{\theta}, \theta_0)\big],$$

respectively, where the infimum on in the first expression is over all estimators $\hat{\theta}$, and in the second expression over all *linear estimators* $\hat{\theta}$, meaning that $\hat{\theta} = Sy$ for a matrix $S \in \mathbb{R}^{n \times n}$. We will also refer to linear estimators as *linear smoothers*. Note that both Laplacian smoothing in (4) and Laplacian eigenmaps in (5) are linear smoothers, but TV denoising in (3) is not. Lastly, in somewhat of an abuse of nomenclature, we will often call the parameter $\theta_0$ in (1) a function, and a set of possible values for $\theta_0$ as in (2) a function space; this comes from thinking of the components of $\theta_0$ as the evaluations of an underlying function over $n$ locations on the grid. This embedding has no formal importance, but it is convenient notationally, and matches the notation in nonparametric statistics.

**Review: TV denoising in 1d.** The classical nonparametric statistics literature [13, 12, 18] provides a more or less complete story for estimation under total variation constraints in 1d. See also [26] for a translation of these results to a setting more consistent (notationally) to that in the current paper. Assume that $d = 1$ and $C_n = C > 0$, a constant (not growing with $n$). The results in [12] imply that

$$R(\mathcal{T}_1(C)) \asymp n^{-2/3}. \tag{6}$$

Furthermore, [18] proved that the TV denoiser $\hat{\theta}^{\mathrm{TV}}$ in (3), with $\lambda \asymp n^{1/3}$, satisfies

$$\mathrm{MSE}(\hat{\theta}^{\mathrm{TV}}, \theta_0) = O_{\mathbb{P}}(n^{-2/3}), \tag{7}$$

for all $\theta_0 \in \mathcal{T}_1(C)$, and is thus minimax rate optimal over $\mathcal{T}_1(C)$. (In assessing rates here and throughout, we do not distinguish between convergence in expectation versus convergence in probability.) Wavelet denoising, under various choices of wavelet bases, also achieves the minimax rate. However, many simpler estimators do not. To be more precise, it is shown in [12] that

$$R_L(\mathcal{T}_1(C)) \asymp n^{-1/2}. \tag{8}$$

Therefore, a substantial number of commonly used nonparametric estimators—such as running mean estimators, smoothing splines, kernel smoothing, Laplacian smoothing, and Laplacian eigenmaps, which are all linear smoothers—have a major deficiency when it comes to estimating functions of bounded variation. Roughly speaking, they will require many more samples to estimate $\theta_0$ within the same degree of accuracy as an optimal method like TV or wavelet denoising (on the order of $\epsilon^{-1/2}$ times more samples to achieve an MSE of $\epsilon$). Further theory and empirical examples (e.g., [11, 12, 26]) offer the following perspective: linear smoothers cannot cope with functions in $T(C)$ that have spatially inhomogeneous smoothness, i.e., that vary smoothly at some locations and vary wildly at others. Linear smoothers can only produce estimates that are smooth throughout, or wiggly throughout, but not a mix of the two. They can hence perform well over smaller, more homogeneous function classes like Sobolev or Holder classes, but not larger ones like total variation classes (or more generally, Besov and Triebel classes), and for these, one must use more sophisticated, nonlinear techniques. A motivating question: does such a gap persist in higher dimensions, between optimal nonlinear and linear estimators, and if so, how big is it?

**Review: TV denoising in multiple dimensions.** Recently, [29] established rates for TV denoising over various graph models, including grids, and [16] made improvements, particularly in the case of $d$-dimensional grids with $d \geq 2$. We can combine Propositions 4 and 6 of [16] with Theorem 3 of [29] to give the following result: if $d \geq 2$, and $C_n$ is an arbitrary sequence (potentially unbounded with $n$), then the TV denoiser $\hat{\theta}^{\mathrm{TV}}$ in (3) satisfies, over all $\theta_0 \in \mathcal{T}_d(C_n)$,

$$\mathrm{MSE}(\hat{\theta}^{\mathrm{TV}}, \theta_0) = O_{\mathbb{P}}\left(\frac{C_n \log n}{n}\right) \text{ for } d = 2, \text{ and } \mathrm{MSE}(\hat{\theta}^{\mathrm{TV}}, \theta_0) = O_{\mathbb{P}}\left(\frac{C_n \sqrt{\log n}}{n}\right) \text{ for } d \geq 3,$$
(9)

with $\lambda \asymp \log n$ for $d = 2$, and $\lambda \asymp \sqrt{\log n}$ for $d \geq 3$. Note that, at first glance, this is a very different result from the 1d case. We expand on this next.

## 2 Summary of results

**A gap in multiple dimensions.** For estimation of $\theta_0$ in (1) when $d \geq 2$, consider, e.g., the simplest possible linear smoother: the mean estimator, $\hat{\theta}^{\mathrm{mean}} = \bar{y}\mathbb{1}$ (where $\mathbb{1} = (1, \ldots, 1) \in \mathbb{R}^n$, the vector of all 1s). Lemma 4, given below, implies that over $\theta_0 \in \mathcal{T}_d(C_n)$, the MSE of the mean estimator is bounded in probability by $C_n^2 \log n / n$ for $d = 2$, and $C_n^2 / n$ for $d \geq 3$. Compare this to (9). When $C_n = C > 0$ is a constant, i.e., when the TV of $\theta_0$ is assumed to be bounded (which is assumed for the 1d results in (6), (7), (8)), this means that the TV denoiser and the mean estimator converge to $\theta_0$ *at the same rate*, basically (ignoring log terms), the "parametric rate" of $1/n$, for estimating a finite-dimensional parameter! That TV denoising and such a trivial linear smoother perform comparably over 2d and 3d grids could not be farther from the story in 1d, where TV denoising is separated by an unbridgeable gap from *all* linear smoothers, as shown in (6), (7), (8).

Our results in Section 3 clarify this conundrum, and can be summarized by three points.

- We argue in Section 3.1 that there is a proper "canonical" scaling for the TV class defined in (2). E.g., when $d = 1$, this yields $C_n \asymp 1$, a constant, but when $d = 2$, this yields $C_n \asymp \sqrt{n}$, and $C_n$ also diverges with $n$ for all $d \geq 3$. Sticking with $d = 2$ as an interesting example, we see that under such a scaling, the MSE rates achieved by TV denoising and the mean estimator respectively, are drastically different; ignoring log terms, these are

$$\frac{C_n}{n} \asymp \frac{1}{\sqrt{n}} \quad \text{and} \quad \frac{C_n^2}{n} \asymp 1,$$
(10)

  respectively. Hence, TV denoising has an MSE rate of $1/\sqrt{n}$, in a setting where the mean estimator has a *constant* rate, i.e., a setting where it is not even known to be consistent.

- We show in Section 3.3 that our choice to study the mean estimator here is not somehow "unlucky" (it is not a particularly bad linear smoother, nor is the upper bound on its MSE loose): the minimax linear risk over $\mathcal{T}_d(C_n)$ is on the order $C_n^2 / n$, for all $d \geq 2$. Thus, even the best linear smoothers have the same poor performance as the mean over $\mathcal{T}_d(C_n)$.

- We show in Section 3.2 that the TV estimator is (essentially) minimax optimal over $\mathcal{T}_d(C_n)$, as the minimax risk over this class scales as $C_n / n$ (ignoring log terms).

To summarize, these results reveal a significant gap between linear smoothers and optimal estimators like TV denoising, for estimation over $\mathcal{T}_d(C_n)$ in $d$ dimensions, with $d \geq 2$, as long as $C_n$ scales appropriately. Roughly speaking, the TV classes encompass a challenging setting for estimation because they are very broad, containing a wide array of functions—both globally smooth functions, said to have homogeneous smoothness, and functions with vastly different levels of smoothness at different grid locations, said to have heterogeneous smoothness. Linear smoothers cannot handle heterogeneous smoothness, and only nonlinear methods can enjoy good estimation properties over the entirety of $\mathcal{T}_d(C_n)$. To reiterate, a telling example is $d = 2$ with the canonical scaling $C_n \asymp \sqrt{n}$, where we see that TV denoising achieves the optimal $1/\sqrt{n}$ rate (up to log factors), meanwhile, the best linear smoothers have max risk that is constant over $\mathcal{T}_2(\sqrt{n})$. See Figure 2 for an illustration.

**Minimax rates over smaller function spaces, and adaptivity.** Sections 4 and 5 are focused on different function spaces, discrete Sobolev spaces, which are $\ell_2$ analogs of discrete TV spaces as we have defined them in (2). Under the canonical scaling of Section 3.1, Sobolev spaces are contained in

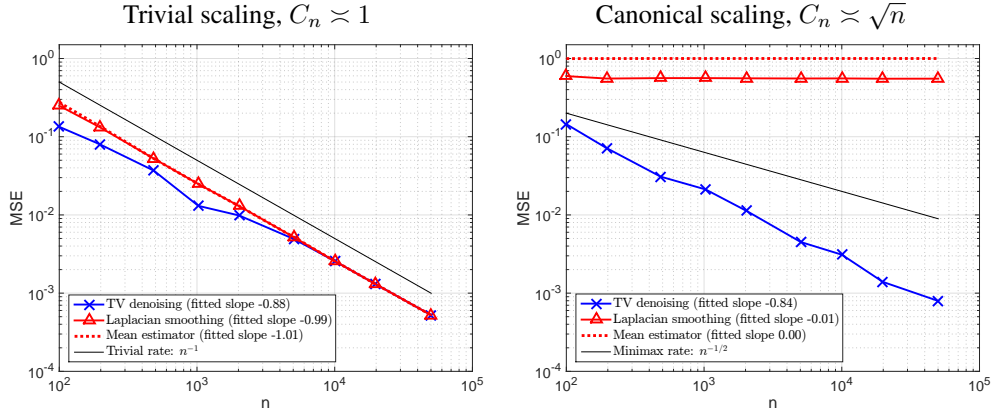

**Figure 2:** *MSE curves for estimation over a 2d grid, under two very different scalings of $C_n$: constant and $\sqrt{n}$. The parameter $\theta_0$ was a "one-hot" signal, with all but one component equal to 0. For each $n$, the results were averaged over 5 repetitions, and Laplacian smoothing and TV denoising were tuned for optimal average MSE.*

TV spaces, and the former can be roughly thought of as containing functions of more homogeneous smoothness. The story now is more optimistic for linear smoothers, and the following is a summary.

- In Section 4, we derive minimax rates for Sobolev spaces, and prove that linear smoothers—in particular, Laplacian smoothing and Laplacian eigenmaps—are optimal over these spaces.

- In Section 5, we discuss an interesting phenomenon, a phase transition of sorts, at $d = 3$ dimensions. When $d = 1$ or 2, the minimax rates for a TV space and its inscribed Sobolev space match; when $d \geq 3$, they do not, and the inscribed Sobolev space has a faster minimax rate. Aside from being an interesting statement about the TV and Sobolev function spaces in high dimensions, this raises an important question of adaptivity over the smaller Sobolev function spaces. As the minimax rates match for $d = 1$ and 2, any method optimal over TV spaces in these dimensions, such as TV denoising, is automatically optimal over the inscribed Sobolev spaces. But the question remains open for $d \geq 3$—does, e.g., TV denoising adapt to the faster minimax rate over Sobolev spaces? We present empirical evidence to suggest that this may be true, and leave a formal study to future work.

**Other considerations and extensions.** There are many problems related to the one that we study in this paper. Clearly, minimax rates for the TV and Sobolev classes over general graphs, not just $d$-dimensional grids, are of interest. Our minimax lower bounds for TV classes actually apply to generic graphs with bounded max degree, though it is unclear whether to what extent they are sharp beyond grids; a detailed study will be left to future work. Another related topic is that of higher-order smoothness classes, e.g., classes containing functions whose *derivatives* are of bounded variation. The natural extension of TV denoising here is called *trend filtering*, defined via the regularization of discrete higher-order derivatives. In the 1d setting, minimax rates, the optimality of trend filtering, and the suboptimality of linear smoothers is already well-understood [26]. Trend filtering has been defined and studied to some extent on general graphs [29], but no notions of optimality have been investigated beyond 1d. This will also be left to future work. Lastly, it is worth mentioning that there are other estimators (i.e., other than the ones we study in detail) that attain or nearly attain minimax rates over various classes we consider in this paper. E.g., wavelet denoising is known to be optimal over TV classes in 1d [12]; and comparing recent upper bounds from [19, 16] with the lower bounds in this work, we see that wavelet denoising is also nearly minimax in 2d (ignoring log terms).

## 3  Analysis over TV classes

### 3.1  Canonical scalings for TV and Sobolev classes

We start by establishing what we call a "canonical" scaling for the radius $C_n$ of the TV ball $\mathcal{T}_d(C_n)$ in (2), as well as the radius $C'_n$ of the Sobolev ball $\mathcal{S}_d(C'_n)$, defined as

$$\mathcal{S}_d(C'_n) = \left\{ \theta : \|D\theta\|_2 \leq C'_n \right\}. \tag{11}$$

Proper scalings for $C_n, C'_n$ will be critical for properly interpreting our new results in $d$ dimensions, in a way that is comparable to known results for $d = 1$ (which are usually stated in terms of the 1d scalings $C_n \asymp 1$, $C'_n \asymp 1/\sqrt{n}$). To study (2), (11), it helps to introduce a third function space,

$$\mathcal{H}_d(1) = \left\{ \theta : \theta_i = f(i_1/\ell \dots, i_d/\ell), \ i = 1, \dots, n, \ \text{for some } f \in \mathcal{H}_d^{\text{cont}}(1) \right\}. \qquad (12)$$

Above, we have mapped each location $i$ on the grid to a multi-index $(i_1, \dots, i_d) \in \{1, \dots, \ell\}^d$, where $\ell = n^{1/d}$, and $\mathcal{H}_d^{\text{cont}}(1)$ denotes the (usual) continuous Holder space on $[0,1]^d$, i.e., functions that are 1-Lipschitz with respect to the $\ell_\infty$ norm. We seek an embedding that is analogous to the embedding of continuous Holder, Sobolev, and total variation spaces in 1d functional analysis, namely,

$$\mathcal{H}_d(1) \subseteq \mathcal{S}_d(C'_n) \subseteq \mathcal{T}_d(C_n). \qquad (13)$$

Our first lemma provides a choice of $C_n, C'_n$ that makes the above true. Its proof, as with all proofs in this paper, can be found in the supplementary document.

**Lemma 1.** *For $d \geq 1$, the embedding in (13) holds with choices $C_n \asymp n^{1-1/d}$ and $C'_n \asymp n^{1/2-1/d}$. Such choices are called the* canonical scalings *for the function classes in (2), (11).*

As a sanity check, both the (usual) continuous Holder and Sobolev function spaces in $d$ dimensions are known to have minimax risks that scale as $n^{-2/(2+d)}$, in a standard nonparametric regression setup (e.g., [14]). Under the canonical scaling $C'_n \asymp n^{1/2-1/d}$, our results in Section 4 show that the discrete Sobolev class $\mathcal{S}_d(n^{1/2-1/d})$ also admits a minimax rate of $n^{-2/(2+d)}$.

### 3.2 Minimax rates over TV classes

The following is a lower bound for the minimax risk of the TV class $\mathcal{T}_d(C_n)$ in (2).

**Theorem 2.** *Assume $n \geq 2$, and denote $d_{\max} = 2d$. Then, for constants $c > 0$, $\rho_1 \in (2.34, 2.35)$,*

$$R(\mathcal{T}_d(C_n)) \geq c \cdot \begin{cases} \dfrac{\sigma C_n \sqrt{1 + \log(\sigma d_{\max} n/C_n)}}{d_{\max} n} & \text{if } C_n \in [\sigma d_{\max}\sqrt{\log n}, \sigma d_{\max} n/\sqrt{\rho_1}] \\ C_n^2/(d_{\max}^2 n) \vee \sigma^2/n & \text{if } C_n < \sigma d_{\max}\sqrt{\log n} \\ \sigma^2/\rho_1 & \text{if } C_n > \sigma d_{\max} n/\sqrt{\rho_1} \end{cases}. \qquad (14)$$

The proof uses a simplifying reduction of the TV class, via $\mathcal{T}_d(C_n) \supseteq B_1(C_n/d_{\max})$, the latter set denoting the $\ell_1$ ball of radius $C_n/d_{\max}$ in $\mathbb{R}^n$. It then invokes a sharp characterization of the minimax risk in normal means problems over $\ell_p$ balls due to [6]. Several remarks are in order.

**Remark 1.** *The first line on the right-hand side in (14) often provides the most useful lower bound. To see this, recall that under the canonical scaling for TV classes, we have $C_n = n^{1-1/d}$. For all $d \geq 2$, this certainly implies $C_n \in [\sigma d_{\max}\sqrt{\log n}, \sigma d_{\max} n/\sqrt{\rho_1}]$, for large $n$.*

**Remark 2.** *Even though its construction is very simple, the lower bound on the minimax risk in (14) is sharp or nearly sharp in many interesting cases. Assume that $C_n \in [\sigma d_{\max}\sqrt{\log n}, \sigma d_{\max} n/\sqrt{\rho_1}]$. The lower bound rate is $C_n \sqrt{\log(n/C_n)}/n$. When $d = 2$, we see that this is very close to the upper bound rate of $C_n \log n/n$ achieved by the TV denoiser, as stated in (9). These two differ by at most a $\log n$ factor (achieved when $C_n \asymp n$). When $d \geq 3$, we see that the lower bound rate is even closer to the upper bound rate of $C_n \sqrt{\log n}/n$ achieved by the TV denoiser, as in (9). These two now differ by at most a $\sqrt{\log n}$ factor (again achieved when $C_n \asymp n$). We hence conclude that the TV denoiser is essentially minimax optimal in all dimensions $d \geq 2$.*

**Remark 3.** *When $d = 1$, and (say) $C_n \asymp 1$, the lower bound rate of $\sqrt{\log n}/n$ given by Theorem 2 is not sharp; we know from [12] (recall (6)) that the minimax rate over $\mathcal{T}_1(1)$ is $n^{-2/3}$. The result in the theorem (and also Theorem 3) in fact holds more generally, beyond grids: for an arbitrary graph $G$, its edge incidence matrix $D$, and $\mathcal{T}_d(C_n)$ as defined in (2), the result holds for $d_{\max}$ equal to the max degree of $G$. It is unclear to what extent this is sharp, for different graph models.*

### 3.3 Minimax linear rates over TV classes

We now turn to a lower bound on the minimax linear risk of the TV class $\mathcal{T}_d(C_n)$ in (2).

**Theorem 3.** *Recall the notation $d_{\max} = 2d$. Then*

$$R_L(\mathcal{T}_d(C_n)) \geq \frac{\sigma^2 C_n^2}{C_n^2 + \sigma^2 d_{\max}^2 n} \vee \frac{\sigma^2}{n} \geq \frac{1}{2}\left( \frac{C_n^2}{d_{\max}^2 n} \wedge \sigma^2 \right) \vee \frac{\sigma^2}{n}. \qquad (15)$$

The proof relies on an elegant meta-theorem on minimax rates from [13], which uses the concept of a "quadratically convex" set, whose minimax linear risk is the same as that of its hardest rectangular subproblem. An alternative proof can be given entirely from first principles.

**Remark 4.** *When $C_n^2$ grows with $n$, but not too fast (scales as $\sqrt{n}$, at most), the lower bound rate in (15) will be $C_n^2/n$. Compared to the $C_n/n$ minimax rate from Theorem 2 (ignoring log terms), we see a clear gap between optimal nonlinear and linear estimators. In fact, under the canonical scaling $C_n \asymp n^{1-1/d}$, for any $d \geq 2$, this gap is seemingly huge: the lower bound for the minimax linear rate will be a constant, whereas the minimax rate from Theorem 2 (ignoring log terms) will be $n^{-1/d}$.*

We now show that the lower bound in Theorem 3 is essentially tight, and remarkably, it is certified by analyzing two trivial linear estimators: the mean estimator and the identity estimator.

**Lemma 4.** *Let $M_n$ denote the largest column norm of $D^\dagger$. For the mean estimator $\hat{\theta}^{\mathrm{mean}} = \bar{y}\mathbb{1}$,*

$$\sup_{\theta_0 \in \mathcal{T}_d(C_n)} \mathbb{E}\big[\mathrm{MSE}(\hat{\theta}^{\mathrm{mean}}, \theta_0)\big] \leq \frac{\sigma^2 + C_n^2 M_n^2}{n},$$

*From Proposition 4 in [16], we have $M_n = O(\sqrt{\log n})$ when $d = 2$ and $M_n = O(1)$ when $d \geq 3$.*

The risk of the identity estimator $\hat{\theta}^{\mathrm{id}} = y$ is clearly $\sigma^2$. Combining this logic with Lemma 4 gives the upper bound $R_L(\mathcal{T}_d(C_n)) \leq (\sigma^2 + C_n^2 M_n^2)/n \wedge \sigma^2$. Comparing this with the lower bound described in Remark 4, we see that the two rates basically match, modulo the $M_n^2$ factor in the upper bound, which only provides an extra $\log n$ factor when $d = 2$. The takeaway message: in the sense of max risk, the best linear smoother does not perform much better than the trivial estimators.

Additional empirical experiments, similar to those shown in Figure 2, are given in the supplement.

# 4 Analysis over Sobolev classes

Our first result here is a lower bound on the minimax risk of the Sobolev class $\mathcal{S}_d(C_n')$ in (11).

**Theorem 5.** *For a universal constant $c > 0$,*

$$R(\mathcal{S}_d(C_n')) \geq \frac{c}{n}\Big((n\sigma^2)^{\frac{2}{d+2}}(C_n')^{\frac{2d}{d+2}} \wedge n\sigma^2 \wedge n^{2/d}(C_n')^2\Big) + \frac{\sigma^2}{n}.$$

Elegant tools for minimax analysis from [13], which leverage the fact that the ellipsoid $\mathcal{S}_d(C_n')$ is orthosymmetric and quadratically convex (after a rotation), are used to prove the result.

The next theorem gives upper bounds, certifying that the above lower bound is tight, and showing that Laplacian eigenmaps and Laplacian smoothing, both linear smoothers, are optimal over $\mathcal{S}_d(C_n')$.

**Theorem 6.** *For Laplacian eigenmaps, $\hat{\theta}^{\mathrm{LE}}$ in (5), with $k \asymp ((n(C_n')^d)^{2/(d+2)} \vee 1) \wedge n$, we have*

$$\sup_{\theta_0 \in \mathcal{S}_d(C_n')} \mathbb{E}\big[\mathrm{MSE}(\hat{\theta}^{\mathrm{LE}}, \theta_0)\big] \leq \frac{c}{n}\Big((n\sigma^2)^{\frac{2}{d+2}}(C_n')^{\frac{2d}{d+2}} \wedge n\sigma^2 \wedge n^{2/d}(C_n')^2\Big) + \frac{c\sigma^2}{n},$$

*for a universal constant $c > 0$, and $n$ large enough. When $d = 1, 2$, or $3$, the same bound holds for Laplacian smoothing $\hat{\theta}^{\mathrm{LS}}$ in (5), with $\lambda \asymp (n/(C_n')^2)^{2/(d+2)}$ (and a possibly different constant $c$).*

# 5 A phase transition, and adaptivity

The TV and Sobolev classes in (2) and (11), respectively, display a curious relationship. We reflect on Theorems 2 and 5, using, for concreteness, the canonical scalings $C_n \asymp n^{1-1/d}$ and $C_n' \asymp n^{1/2-1/d}$ (that, recall, guarantee $\mathcal{S}_d(C_n') \subseteq \mathcal{T}_d(C_n)$). When $d = 1$, both the TV and Sobolev classes have a minimax rate of $n^{-2/3}$ (this TV result is actually due to [12], as stated in (6), not Theorem 2). When $d = 2$, both the TV and Sobolev classes again have the same minimax rate of $n^{-1/2}$, the caveat being that the rate for TV class has an extra $\sqrt{\log n}$ factor. But for all $d \geq 3$, the rates for the canonical TV and Sobolev classes differ, and the smaller Sobolev spaces have faster rates than their inscribing TV spaces. This may be viewed as a phase transition at $d = 3$; see Table 1.

We may paraphrase to say that 2d is just like 1d, in that expanding the Sobolev ball into a larger TV ball does not hurt the minimax rate, and methods like TV denoising are automatically *adaptive*, i.e.,

| Function class | Dimension 1 | Dimension 2 | Dimension $d \geq 3$ |
|---|---|---|---|
| TV ball $\mathcal{T}_d(n^{1-1/d})$ | $n^{-2/3}$ | $n^{-1/2}\sqrt{\log n}$ | $n^{-1/d}\sqrt{\log n}$ |
| Sobolev ball $\mathcal{S}_d(n^{1/2-1/d})$ | $n^{-2/3}$ | $n^{-1/2}$ | $n^{-\frac{2}{2+d}}$ |

Table 1: *Summary of rates for canonically-scaled TV and Sobolev spaces.*

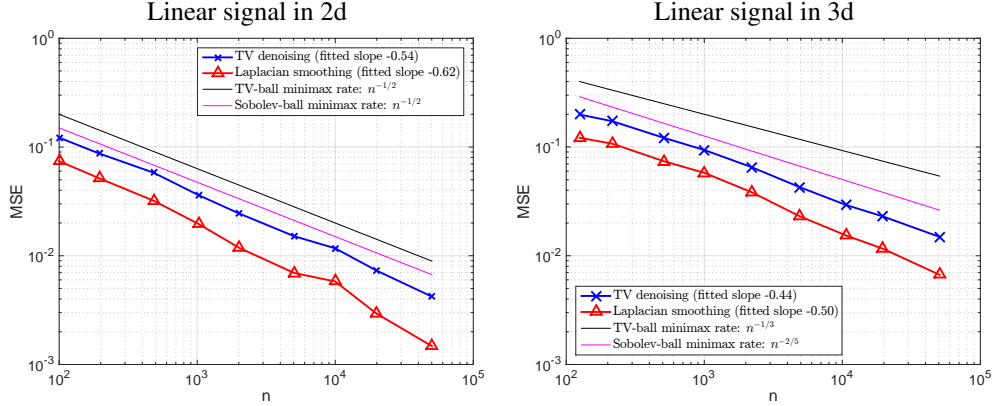

Figure 3: *MSE curves for estimating a "linear" signal, a very smooth signal, over 2d and 3d grids. For each $n$, the results were averaged over 5 repetitions, and Laplacian smoothing and TV denoising were tuned for best average MSE performance. The signal was set to satisfy $\|D\theta_0\|_2 \asymp n^{1/2-1/d}$, matching the canonical scaling.*

optimal over both the bigger and smaller classes. However, as soon as we enter the 3d world, it is no longer clear whether TV denoising can adapt to the smaller, inscribed Sobolev ball, whose minimax rate is faster, $n^{-2/5}$ versus $n^{-1/3}$ (ignoring log factors). Theoretically, this is an interesting open problem that we do not approach in this paper and leave to future work.

We do, however, investigate the matter empirically: see Figure 3, where we run Laplacian smoothing and TV denoising on a highly smooth "linear" signal $\theta_0$. This is constructed so that each component $\theta_i$ is proportional to $i_1 + i_2 + \ldots + i_d$ (using the multi-index notation $(i_1, \ldots, i_d)$ of (12) for grid location $i$), and the Sobolev norm is $\|D\theta_0\|_2 \asymp n^{1/2-1/d}$. Arguably, these are among the "hardest" types of functions for TV denoising to handle. The left panel, in 2d, is a case in which we know that TV denoising attains the minimax rate; the right panel, in 3d, is a case in which we do not, though empirically, TV denoising surely seems to be doing better than the slower minimax rate of $n^{-1/3}$ (ignoring log terms) that is associated with the larger TV ball.

Even if TV denoising is shown to be minimax optimal over the inscribed Sobolev balls when $d \geq 3$, note that this does not necessarily mean that we should scrap Laplacian smoothing in favor of TV denoising, in all problems. Laplacian smoothing is the unique Bayes estimator in a normal means model under a certain Markov random field prior (e.g., [22]); statistical decision theory therefore tells that it is *admissible*, i.e., no other estimator—TV denoising included—can uniformly dominate it.

## 6 Discussion

We conclude with a quote from Albert Einstein: "Everything should be made as simple as possible, but no simpler". In characterizing the minimax rates for TV classes, defined over $d$-dimensional grids, we have shown that simple methods like Laplacian smoothing and Laplacian eigenmaps—or even in fact, all linear estimators—must be passed up in favor of more sophisticated, nonlinear estimators, like TV denoising, if one wants to attain the optimal max risk. Such a result was previously known when $d = 1$; our work has extended it to all dimensions $d \geq 2$. We also characterized the minimax rates over discrete Sobolev classes, revealing an interesting phase transition where the optimal rates over TV and Sobolev spaces, suitably scaled, match when $d = 1$ and 2 but diverge for $d \geq 3$. It is an open question as to whether an estimator like TV denoising can be optimal over both spaces, for all $d$.

**Acknolwedgements.** We thank Jan-Christian Hutter and Philippe Rigollet, whose paper [16] inspired us to think carefully about problem scalings (i.e., radii of TV and Sobolev classes) in the first place. YW was supported by NSF Award BCS-0941518 to CMU Statistics, a grant by Singapore NRF under its International Research Centre @ Singapore Funding Initiative, and a Baidu Scholarship. RT was supported by NSF Grants DMS-1309174 and DMS-1554123.

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
