[Supplementary Material]

# Supplement to "Total Variation Classes Beyond 1d: Minimax Rates, and the Limitations of Linear Smoothers"

**Veeranjaneyulu Sadhanala**
Machine Learning Department
Carnegie Mellon University
Pittsburgh, PA 15213
vsadhana@cs.cmu.edu

**Yu-Xiang Wang**
Machine Learning Department
Carnegie Mellon University
Pittsburgh, PA 15213
yuxiangw@cs.cmu.edu

**Ryan J. Tibshirani**
Department of Statistics
Carnegie Mellon University
Pittsburgh, PA 15213
ryantibs@stat.cmu.edu

In this supplement, we present proofs, additional technical results, and additional simulations for "Total Variation Classes Beyond 1d: Minimax Rates, and the Limitations of Linear Smoothers".

## A.1 Proof of Lemma 1 (canonical scaling)

Suppose that $\theta \in \mathcal{H}_d(1)$ that is a discretization of a 1-Lipschitz function $f$, i.e., $\theta_i = f(i_1/\ell \ldots, i_d/\ell)$, $i = 1, \ldots, n$. We first we compute and bound its squared Sobolev norm

$$
\|D\theta\|_2^2 = \sum_{(i,j)\in E} (\theta_i - \theta_j)^2 = \sum_{(i,j)\in E} \big( f(i_1/\ell, \ldots, i_d/\ell) - f(j_1/\ell, \ldots, j_d/\ell) \big)^2
$$
$$
\leq \sum_{(i,j)\in E} \big\| (i_1/\ell, \ldots, i_d/\ell) - (j_1/\ell, \ldots, j_d/\ell) \big\|_\infty^2
$$
$$
= m/\ell^2,
$$

where, recall, we denote by $m = |E|$ the number of edges in the grid. In the second line we used the 1-Lipschitz property of $f$, and in the third we used that multi-indices corresponding to adjacent locations on the grid are exactly 1 apart, in $\ell_\infty$ distance. Thus we see that setting $C_n' = \sqrt{m}/\ell$ gives the desired containment $\mathcal{S}_d(C_n') \supseteq \mathcal{H}_d(1)$. It is always true that $m \asymp n$ for a $d$-dimensional grid (though the constant may depend on $d$), so that $\sqrt{m}/\ell \asymp n^{1/2-1/d}$. This completes the proof for the Sobolev class scaling.

As for TV class scaling, the result follows from the simple fact that $\|x\|_1 \leq \sqrt{m}\|x\|_2$ for any $x \in \mathbb{R}^m$, so that we may take $C_n = \sqrt{m}C_n' = n^{1-1/d}$. $\qquad\square$

## A.2 Proof of Theorem 2 (minimax rates over TV classes)

Here and henceforth, we use the notation $B_p(r) = \{x : \|x\|_p \leq r\}$ for the $\ell_p$ ball of radius $r$, where $p, r > 0$ (and the ambient dimension will be determined based on the context).

We begin with a very simple lemma, that embeds an $\ell_1$ ball inside the TV ball $\mathcal{T}_d(C_n)$.

**Lemma A.1.** *Let $G$ be a graph with maximum degree $d_{\max}$, and let $D \in \mathbb{R}^{m \times n}$ be its incidence matrix. Then for any $r > 0$, it holds that $B_1(r/d_{\max}) \subseteq T_d(r)$.*

*Proof.* Write $D_i$ for the $i$th column of $D$. The proof follows from the observation that, for any $\theta$,

$$
\|D\theta\|_1 = \left\| \sum_{i=1}^n D_i\theta_i \right\|_1 \leq \sum_{i=1}^n \|D_i\|_1|\theta_i| \leq \Big( \max_{i=1,\ldots,n} \|D_i\|_1 \Big)\|\theta\|_1 = d_{\max}\|\theta\|_1.
$$

$\qquad\square$

To prove Theorem 2, we will rely on a result from Birge and Massart [1], which gives a lower bound for the risk in a normal means problem, over $\ell_p$ balls. Another related, earlier result is that of Donoho and Johnstone [3]; however, the Birge and Massart result places no restrictions on the radius of the ball in question, whereas the Donoho and Johnstone result does. Translated into our notation, the Birge and Massart result is as follows.

**Lemma A.2** (Proposition 5 of Birge and Massart [1]). *Assume i.i.d. observations $y_i \sim N(\theta_{0,i}, \sigma^2)$, $i = 1, \ldots, n$, and $n \geq 2$. Then the minimax risk over the $\ell_p$ ball $B_p(r_n)$, where $0 < p < 2$, satisfies*

$$n \cdot R(B_p(r_n)) \geq c \cdot \begin{cases} \sigma^{2-p} r_n^p \left[ 1 + \log\left( \frac{\sigma^p n}{r_n^p} \right) \right]^{1-p/2} & \text{if } \sigma\sqrt{\log n} \leq r_n \leq \sigma n^{1/p}/\sqrt{\rho_p} \\ r_n^2 & \text{if } r_n < \sigma\sqrt{\log n} \\ \sigma^2 n/\rho_p & \text{if } r_n > \sigma n^{1/p}/\sqrt{\rho} \end{cases}.$$

*Here $c > 0$ is a universal constant, and $\rho_p > 1.76$ is the unique solution of $\rho_p \log \rho_p = 2/p$.*

Finally, applying Lemma A.2 to $B_1(C_n/d_{\max})$ almost gives the lower bound as stated in Theorem 2. However, note that the minimax risk in question is trivially lower bounded by $\sigma^2/n$, because

$$\inf_{\hat{\theta}} \sup_{\theta_0 \in \mathcal{T}_d(C_n)} \frac{1}{n} \mathbb{E}\|\hat{\theta} - \theta_0\|_2^2 \geq \inf_{\hat{\theta}} \sup_{\theta_0 : \theta_{0,1} = \ldots = \theta_{0,n}} \frac{1}{n} \sum_{i=1}^n \mathbb{E}(\hat{\theta}_i - \theta_{0,1})^2$$

$$= \inf_{\hat{\theta}_1} \sup_{\theta_{0,1}} \mathbb{E}(\hat{\theta}_1 - \theta_{0,1})^2$$

$$= \frac{\sigma^2}{n}.$$

In the second to last line, the problem is to estimate a 1-dimensional mean parameter $\theta_{0,1}$, given the observations $y_i \sim N(\theta_{0,1}, \sigma^2)$, i.i.d., for $i = 1, \ldots, n$; this has a well-known minimax risk of $\sigma^2/n$. What this means for our TV problem: to derive a lower bound for the minimax rate over $\mathcal{T}_d(C_n)$, we may take the maximum of the result of applying Lemma A.2 to $B_1(C_n/d_{\max})$ and $\sigma^2/n$. One can see that the term $\sigma^2/n$ only plays a role for small $C_n$, i.e., it effects the case when $C_n < \sigma d_{\max}\sqrt{\log n}$, where the lower bound becomes $C_n^2/(d_{\max}^2 n) \vee \sigma^2/n$. □

## A.3 Proof of Theorem 3 (minimax linear rates over TV classes)

First we recall a few definitions, from Donoho et al. [4]. Given a set $A \subseteq \mathbb{R}^k$, its *quadratically convex hull* $\mathrm{qconv}(A)$ is defined as

$$\mathrm{qconv}(A) = \{(x_1, \ldots, x_k) : (x_1^2, \ldots, x_k^2) \in \mathrm{conv}(A_+^2)\}, \quad \text{where}$$

$$A_+^2 = \{(a_1^2, \ldots, a_k^2) : a \in A, \ a_i \geq 0, \ i = 1, \ldots, k\}.$$

(Here $\mathrm{conv}(B)$ denotes the convex hull of a set $B$.) Furthermore, the set $A$ is called *quadratically convex* provided that $\mathrm{qconv}(A) = A$. Also, $A$ is called *orthosymmetric* provided that $(a_1, \ldots, a_k) \in A$ implies $(\sigma_1 a_1, \ldots, \sigma_k a_k) \in A$, for any choice of signs $\sigma_1, \ldots, \sigma_k \in \{-1, 1\}$.

Now we proceed with the proof. Following from equation (7.2) of Donoho et al. [4],

$$\mathrm{qconv}\big(B_1(C_n/d_{\max})\big) = B_2(C_n/d_{\max}).$$

Theorem 11 of Donoho et al. [4] states that, for orthosymmetric, compact sets, such as $B_1(C_n/d_{\max})$, the minimax linear risk equals that of its quadratically convex hull. Moreover, Theorem 7 of Donoho et al. [4] tells us that for sets that are orthosymmetric, compact, convex, and quadratically convex, such as $B_2(C_n/d_{\max})$, the minimax linear risk is the same as the minimax linear risk over the worst rectangular subproblem. We consider $B_\infty(C_n/(d_{\max}\sqrt{n}))$, and abbreviate $r_n = C_n/(d_{\max}\sqrt{n})$. It is fruitful to study rectangles because the problem separates across dimensions, as in

$$\inf_{\hat{\theta} \text{ linear}} \sup_{\theta_0 \in B_\infty(r_n)} \mathbb{E}\left[ \frac{1}{n} \sum_{i=1}^n (\hat{\theta}_i - \theta_{0,i})^2 \right] = \frac{1}{n} \sum_{i=1}^n \left[ \inf_{\hat{\theta}_i \text{ linear}} \sup_{|\theta_{0,i}| \leq r_n} \mathbb{E}(\hat{\theta}_i - \theta_{0,i})^2 \right]$$

$$= \inf_{\hat{\theta}_1 \text{ linear}} \sup_{|\theta_{0,1}| \leq r_n} \mathbb{E}(\hat{\theta}_1 - \theta_{0,1})^2.$$

Thus it suffices to compute the minimax linear risk over the 1d class $\{\theta_{0,1} : |\theta_{0,1}| \leq r_n\}$. It is easily shown (e.g., see Section 2 of Donoho et al. [4]) that this is $r_n^2 \sigma^2 / (r_n^2 + \sigma_2^2)$, and so this is precisely the minimax linear risk for $B_2(C_n / d_{\max})$, and for $B_1(C_n / d_{\max})$.

To get the first lower bound as stated in the theorem, we simply take a maximum of $r_n^2 \sigma^2 / (r_n^2 + \sigma_2^2)$ and $\sigma^2 / n$, as the latter is the minimax risk for estimating a 1-dimensional mean parameter given $n$ observations in a normal model with variance $\sigma^2$, recall the end of the proof of Theorem 2. To get the second, we use the fact that $2ab / (a + b) \geq \min\{a, b\}$. This completes the proof. $\qquad \square$

## A.4 Alternative proof of Theorem 3

Here, we reprove Theorem 3 using elementary arguments. We write $y = \theta_0 + \epsilon$, for $\epsilon \sim N(0, \sigma^2 I)$. Given an arbitrary linear estimator, $\hat{\theta} = Sy$ for a matrix $S \in \mathbb{R}^{n \times n}$, observe that

$$
\begin{aligned}
\mathbb{E}\big[\mathrm{MSE}(\hat{\theta}, \theta_0)\big] = \frac{1}{n}\mathbb{E}\|\hat{\theta} - \theta_0\|_2^2 &= \frac{1}{n}\mathbb{E}\|S(\theta_0 + \epsilon) - \theta_0\|_2^2 \\
&= \frac{1}{n}\mathbb{E}\|S\epsilon\|_2^2 + \frac{1}{n}\|(S - I)\theta_0\|_2^2 \\
&= \frac{\sigma^2}{n}\|S\|_F^2 + \frac{1}{n}\|(S - I)\theta_0\|_2^2, \qquad (\mathrm{A.1})
\end{aligned}
$$

which we may view as the variance and (squared) bias terms, respectively. Now denote by $e_i$ the $i$th standard basis vector, and consider

$$
\begin{aligned}
\frac{\sigma^2}{n}\|S\|_F^2 + \left( \sup_{\theta_0 : \|D\theta_0\|_1 \leq C_n} \frac{1}{n}\|(S - I)\theta_0\|_2^2 \right) &\geq \frac{\sigma^2}{n}\|S\|_F^2 + \frac{C_n^2}{d_{\max}^2 n}\left( \max_{i=1,\dots,n} \|(I - S)e_i\|_2^2 \right) \\
&\geq \frac{\sigma^2}{n}\|S\|_F^2 + \frac{C_n^2}{d_{\max}^2 n^2}\sum_{i=1}^n \|(I - S)e_i\|_2^2 \\
&= \frac{\sigma^2}{n}\|S\|_F^2 + \frac{C_n^2}{d_{\max}^2 n^2}\|(I - S)\|_F^2 \\
&\geq \frac{\sigma^2}{n}\sum_{i=1}^n S_{ii}^2 + \frac{C_n^2}{d_{\max}^2 n^2}\sum_{i=1}^n (1 - S_{ii})^2 \\
&= \frac{1}{n}\sum_{i=1}^n \left( \sigma^2 S_{ii}^2 + \frac{C_n^2}{d_{\max}^2 n}(1 - S_{ii})^2 \right).
\end{aligned}
$$

Here $S_{ii}$, $i = 1, \dots, n$ denote the diagonal entries of $S$. To bound each term in the sum, we apply the simple inequality $ax^2 + b(1 - x)^2 \geq ab/(a + b)$ for all $x$ (since a short calculation shows that the quadratic in $x$ here is minimized at $x = b/(a + b)$). We may continue on lower bounding the last displayed expression, giving

$$
\frac{\sigma^2}{n}\|S\|_F^2 + \left( \sup_{\theta_0 : \|D\theta_0\|_1 \leq C_n} \frac{1}{n}\|(S - I)\theta_0\|_2^2 \right) \geq \frac{\sigma^2 C_n^2}{C_n^2 + \sigma^2 d_{\max}^2 n}.
$$

Lastly, we may take the maximum of this with $\sigma^2 / n$ in order to derive a final lower bound, as argued in the proof of Theorem 3. $\qquad \square$

## A.5 Proof of Lemma 4 (mean estimator over TV classes)

For this estimator, the smoother matrix is $S = \mathbb{1}\mathbb{1}^T / n$ and so $\|S\|_F^2 = 1$. From (A.1), we have

$$
\mathbb{E}\big[\mathrm{MSE}(\hat{\theta}^{\mathrm{mean}}, \theta_0)\big] = \frac{\sigma^2}{n} + \frac{1}{n}\|\theta_0 - \bar{\theta}_0 \mathbb{1}\|_2^2,
$$

where $\bar{\theta}_0 = (1/n)\sum_{i=1}^{n} \theta_{0,i}$. Now

$$\sup_{\theta_0 : \|D\theta_0\|_1 \le C_n} \frac{1}{n}\|\theta_0 - \bar{\theta}_0 \mathbb{1}\|_2^2 = \sup_{x \in \text{row}(D) : \|Dx\|_1 \le C_n} \frac{1}{n}\|x\|_2^2$$

$$= \sup_{z \in \text{col}(D) : \|z\|_1 \le C_n} \frac{1}{n}\|D^\dagger z\|_2^2$$

$$\le \sup_{z : \|z\|_1 \le C_n} \frac{1}{n}\|D^\dagger z\|_2^2$$

$$= \frac{C_n^2}{n} \max_{i=1,\dots,n} \|D_i^\dagger\|_2^2$$

$$\le \frac{C_n^2 M_n^2}{n},$$

which establishes the desired bound. $\qquad\square$

## A.6 Additional experiments comparing TV denoising and Laplacian smoothing

Figure A.1: *MSE curves for estimating a "piecewise constant" signal, having a single elevated region, over 2d and 3d grids. For each $n$, the results were averaged over 5 repetitions, and the Laplacian smoothing and TV denoising estimators were tuned for best average MSE performance. We set $\theta_0$ to satisfy $\|D\theta_0\|_1 \asymp n^{1-1/d}$, matching the canonical scaling. Note that all estimators achieve better performance than that dictated by their minimax rates.*

## A.7 Proof of Theorem 5 (minimax rates over Sobolev classes)

Recall that we denote by $L = V\Sigma V^T$ the eigendecomposition of the graph Laplacian $L = D^T D$, where $\Sigma = \text{diag}(\rho_1, \dots, \rho_n)$ with $0 = \rho_1 < \rho_2 \le \dots \le \rho_n$, and where $V \in \mathbb{R}^{n \times n}$ has orthonormal columns. Also denote by $D = U\Sigma^{1/2}V^T$ the singular value decomposition of the edge incidence matrix $D$, where $U \in \mathbb{R}^{m \times n}$ has orthonormal columns.[1] First notice that

$$\|D\theta_0\|_2 = \|U\Sigma^{1/2}V^T\theta_0\|_2 = \|\Sigma^{1/2}V^T\theta_0\|_2.$$

This suggests that a rotation by $V^T$ will further simplify the minimax risk over $\mathcal{S}_d(C_n')$, i.e.,

$$\inf_{\hat{\theta}} \sup_{\theta_0 : \|\Sigma^{1/2} V^T \theta_0\|_2 \le C_n'} \frac{1}{n} \mathbb{E}\|\hat{\theta} - \theta_0\|_2^2 = \inf_{\hat{\theta}} \sup_{\theta_0 : \|\Sigma^{1/2} V^T \theta_0\|_2 \le C_n'} \frac{1}{n} \mathbb{E}\|V^T\hat{\theta} - V^T\theta_0\|_2^2$$

$$= \inf_{\hat{\gamma}} \sup_{\gamma_0 : \|\Sigma^{1/2}\gamma_0\|_2 \le C_n'} \frac{1}{n} \mathbb{E}\|\hat{\gamma} - \gamma_0\|_2^2, \tag{A.2}$$

where we have rotated and now consider the new parameter $\gamma_0 = V^T\theta_0$, constrained to lie in

$$\mathcal{E}_d(C_n') = \left\{ \gamma : \sum_{i=2}^n \rho_i \gamma_i^2 \le (C_n')^2 \right\}.$$

To be clear, in the rotated setting (A.2) we observe a vector $y' = V^T y \sim N(\gamma_0, \sigma^2 I)$, and the goal is to estimate the mean parameter $\gamma_0$. Since there are no constraints along the first dimension, we can separate out the MSE in (A.2) into that incurred on the first component, and all other components. Decomposing $\gamma_0 = (\alpha_0, \beta_0) \in \mathbb{R}^{1 \times (n-1)}$, with similar notation for an estimator $\hat{\gamma}$,

$$\inf_{\hat{\gamma}} \sup_{\gamma_0 \in \mathcal{E}_d(C_n')} \frac{1}{n} \mathbb{E}\|\hat{\gamma} - \gamma_0\|_2^2 = \inf_{\hat{\alpha}} \sup_{\alpha_0} \frac{1}{n} \mathbb{E}(\hat{\alpha} - \alpha_0)^2 + \inf_{\hat{\beta}} \sup_{\beta_0 \in P_{-1}(\mathcal{E}_d(C_n'))} \frac{1}{n} \mathbb{E}\|\hat{\beta} - \beta_0\|_2^2$$

$$= \frac{\sigma^2}{n} + \inf_{\hat{\beta}} \sup_{\beta_0 \in P_{-1}(\mathcal{E}_d(C_n'))} \frac{1}{n} \mathbb{E}\|\hat{\beta} - \beta_0\|_2^2, \tag{A.3}$$

where $P_{-1}$ projects onto all coordinate axes but the 1st, i.e., $P_{-1}(x) = (0, x_2, \ldots, x_n)$, and in the second line we have used the fact that the minimax risk for estimating a 1-dimensional parameter $\alpha_0$ given an observation $z \sim N(\alpha_0, \sigma^2)$ is simply $\sigma^2$.

Let us lower bound the second term in (A.3), i.e., $R(P_{-1}(\mathcal{E}_d(C_n')))$. The ellipsoid $P_{-1}(\mathcal{E}_d(C_n'))$ is orthosymmetric, compact, convex, and quadratically convex, hence Theorem 7 in Donoho et al. [4] tells us that its minimax linear risk is the minimax linear risk of its hardest rectangular subproblem. Further, Lemma 6 in Donoho et al. [4] then tells us the minimax linear risk of its hardest rectangular subproblem is, up to a constant factor, the same as the minimax (nonlinear) risk of the full problem. More precisely, Lemma 6 and Theorem 7 from Donoho et al. [4] imply

$$R(P_{-1}(\mathcal{E}_d(C_n'))) \ge \frac{4}{5} R_L(P_{-1}(\mathcal{E}_d(C_n'))) = \sup_{H \subseteq P_{-1}(\mathcal{E}_d(C_n'))} R_L(H), \tag{A.4}$$

where the supremum above is taken over all rectangular subproblems, i.e., all rectangles $H$ contained in $P_{-1}(\mathcal{E}_d(C_n'))$.

To study rectangular subproblems, it helps to reintroduce the multi-index notation for a location $i$ on the $d$-dimensional grid, writing this as $(i_1, \ldots, i_d) \in \{1, \ldots, \ell\}^d$, where $\ell = n^{1/d}$. For a parameter $2 \le \tau \le \ell$, we consider rectangular subsets of the form[2]

$$H(\tau) = \left\{ \beta \in \mathbb{R}^{n-1} : |\beta_i| \le t_i(\tau), \ i = 2, \ldots, n \right\}, \quad \text{where}$$

$$t_i(\tau) = \begin{cases} C_n' / (\sum_{j_1, \ldots, j_d \le \tau} \rho_{j_1, \ldots, j_d})^{1/2} & \text{if } i_1, \ldots, i_d \le \tau \\ 0 & \text{otherwise} \end{cases}, \quad \text{for } i = 2, \ldots, n.$$

It is not hard to check that $H(\tau) \subseteq \{\beta \in \mathbb{R}^{n-1} : \sum_{i=2}^n \rho_i \beta_i^2 \le (C_n')^2\} = P_{-1}(\mathcal{E}_d(C_n'))$. Then, from (A.4),

$$R(P_{-1}(\mathcal{E}_d(C_n'))) \ge \sup_{\tau} R_L(H(\tau)) = \sup_{\tau} \frac{1}{n} \sum_{i=1}^n \frac{t_i(\tau)^2 \sigma^2}{t_i(\tau)^2 + \sigma^2}$$

$$= \sup_{\tau} \frac{1}{n} \frac{(\tau^d - 1)\sigma^2 (C_n')^2}{(C_n')^2 + \sum_{j_1, \ldots, j_d \le \tau} \rho_{j_1, \ldots, j_d}}.$$

The first equality is due to the fact that the minimax risk for rectangles decouples across dimensions, and the 1d minimax linear risk is straightforward to compute for an interval, as argued in the proof

Theorem 3; the second equality simply comes from a short calculation following the definition of $t_i(\tau)$, $i = 2, \ldots, n$. Applying Lemma A.3, on the eigenvalues of the graph Laplacian matrix $L$ for a $d$-dimensional grid, we have that for a constant $c > 0$,

$$\frac{(\tau^d - 1)\sigma^2(C_n')^2}{(C_n')^2 + \sum_{j_1, \ldots, j_d \le \tau} \rho_{j_1, \ldots, j_d}} \ge \frac{(\tau^d - 1)\sigma^2(C_n')^2}{(C_n')^2 + c\sigma^2\tau^{d+2}/\ell^2} \ge \frac{1}{2}\frac{\sigma^2(C_n')^2}{(C_n')^2\tau^{-d} + c\sigma^2\tau^2/\ell^2}.$$

We can choose $\tau$ to maximize the expression on the right above, given by

$$\tau^* = \left(\frac{\ell^2(C_n')^2}{c\sigma^2}\right)^{\frac{1}{d+2}}.$$

When $2 \le \tau^* \le \ell$, this provides us with the lower bound on the minimax risk

$$R(P_{-1}(\mathcal{E}_d(C_n'))) \ge R_L(H(\tau^*)) \ge \frac{1}{2n}\frac{\tau^d\sigma^2(C_n')^2}{2(c\sigma^2)^{\frac{d}{d+2}}(C_n')^{\frac{4}{d+2}}\ell^{-\frac{2d}{d+2}}} = \frac{c_1}{n}(n\sigma^2)^{\frac{2}{d+2}}(C_n')^{\frac{2d}{d+2}},$$
(A.5)

for a constant $c_1 > 0$. When $\tau^* < 2$, we can use $\tau = 2$ as lower bound on the minimax risk,

$$R(P_{-1}(\mathcal{E}_d(C_n'))) \ge R_L(H(2)) \ge \frac{1}{2n}\frac{\sigma^2\ell^2(C_n')^2}{\ell^2(C_n')^22^{-d} + c\sigma^22^2} \ge \frac{c_2}{n}\ell^2(C_n')^2,$$
(A.6)

for a constant $c_2 > 0$, where in the last inequality, we used the fact that $\ell^2(C_n')^2 \le c\sigma^22^{d+2}$ (just a constant) since we are in the case $\tau^* < 2$. Finally, when $\tau^* > \ell$, we can use $\tau = \ell$ as a lower bound on the minimax risk,

$$R(P_{-1}(\mathcal{E}_d(C_n'))) \ge R_L(H(\ell)) \ge \frac{1}{2n}\frac{\sigma^2(C_n')^2}{\ell^{-d}(C_n')^2 + c\sigma^2} \ge c_3\sigma^2,$$
(A.7)

for a constant $c_3 > 0$, where in the last inequality, we used that $c\sigma^2 \le \ell^{-d}(C_n')^2$ as we are in the case $\tau^* > \ell$. Taking a minimum of the lower bounds in (A.5), (A.6), (A.7), as a way to navigate the cases, gives us a final lower bound on $R(P_{-1}(\mathcal{E}_d(C_n')))$, and completes the proof.

## A.8 Proof of Theorem 6 (Laplacian eigenmaps and Laplacian smoothing over Sobolev classes)

We will prove the results for Laplacian eigenmaps and Laplacian separately.

**Laplacian eigenmaps.** The smoother matrix for this estimator is $S_k = V_{[k]}V_{[k]}^T$, for a tuning parameter $k = 1, \ldots, n$. From (A.1),

$$\mathbb{E}\big[\mathrm{MSE}(\hat{\theta}^{\mathrm{LE}}, \theta_0)\big] = \frac{\sigma^2}{n}k + \frac{1}{n}\|(I - S_k)\theta_0\|_2^2.$$

Now we write $k = \tau^d$, and analyze the max risk of the second term,

$$\sup_{\theta_0: \|D\theta_0\|_2 \le C_n'} \frac{1}{n}\|(I - S_k)\theta_0\|_2^2 = \sup_{z: \|z\|_2 \le C_n'} \frac{1}{n}\|(I - S_k)D^\dagger z\|_2^2$$

$$= \frac{(C_n')^2}{n}\sigma_{\max}^2\big((I - S_k)D^\dagger\big)$$

$$\le \frac{(C_n')^2}{n}\frac{1}{4\sin^2(\pi\tau/(2\ell))}$$

$$\le \frac{(C_n')^2}{n}\frac{4\ell^2}{\pi^2\tau^2}.$$

Here we denote by $\sigma_{\max}(A)$ the maximum singular value of a matrix $A$. The last inequality above used the simple lower bound $\sin(x) \ge x/2$ for $x \in [0, \pi/2]$. The earlier inequality used that

$$(I - S_k)D^\dagger = (I - V_{[k]}V_{[k]}^T)V^T(\Sigma^\dagger)^{1/2}U^T = [0, \ldots, 0, V_{k+1}, \ldots, V_n](\Sigma^\dagger)^{1/2}U^T,$$

where we have kept the same notation for the singular value decomposition of $D$ as in the proof of Theorem 5. Therefore $\sigma_{\max}^2((I - S_k)D^\dagger)$ is the reciprocal of the $(k + 1)$st smallest eigenvalue $\rho_{k+1}$

of the graph Laplacian $L$. For any subset $A$ of the set of eigenvalues $\lambda(L) = \{\rho_1, \ldots, \rho_n\}$ of the Laplacian, with $|A| = k$, note that $\rho_{k+1} \geq \min \lambda(L) \setminus A$. This means that, for our $d$-dimensional grid,

$$\rho_{k+1} \geq \min \ \lambda(L) \setminus \{\rho_{i_1,\ldots,i_d} : i_1, \ldots, i_d \leq \tau\}$$
$$= 4\sin^2(\pi\tau/(2\ell)),$$

where recall $\ell = n^{1/d}$, as explained by (A.8), in the proof of Lemma A.3.

Hence, we have established

$$\sup_{\theta_0 : \|D\theta_0\|_2 \leq C_n'} \mathbb{E}\big[\mathrm{MSE}(\hat{\theta}^{\mathrm{LE}}, \theta_0)\big] \leq \frac{\sigma^2}{n} + \frac{\sigma^2}{n}\tau^d + \frac{(C_n')^2}{n}\frac{4\ell^2}{\pi^2\tau^2}.$$

Choosing $\tau$ to balance the two terms on the right-hand side above results in $\tau^* = (2\ell C_n'/(\pi\sigma))^{\frac{2}{d+2}}$. Plugging in this choice of $\tau$, while utilizing the bounds $1 \leq \tau \leq \ell$, very similar to the arguments given at the end of the proof of Theorem 5, gives the result for Laplacian eigenmaps.

**Laplacian smoothing.** The smoother matrix for this estimator is $S_\lambda = (I + \lambda L)^{-1}$, for a tuning parameter $\lambda \geq 0$. From (A.1),

$$\mathbb{E}\big[\mathrm{MSE}(\hat{\theta}^{\mathrm{LS}}, \theta_0)\big] = \frac{\sigma^2}{n}\sum_{i=1}^{n}\frac{1}{(1 + \lambda\rho_i)^2} + \frac{1}{n}\|(I - S_\lambda)\theta_0\|_2^2.$$

When $d = 1, 2$, or $3$, the first term upper is bounded by $c_1\sigma^2/n + c_2\sigma^2/\lambda^{d/2}$, for some constants $c_1, c_2 > 0$, by Lemma A.4. As for the second term,

$$\sup_{\theta_0 : \|D\theta_0\|_2 \leq C_n'} \frac{1}{n}\|(I - S_\lambda)\theta_0\|_2^2 = \sup_{z : \|z\|_2 \leq C_n'} \|(I - S_\lambda)D^\dagger z\|_2^2$$
$$= \frac{(C_n')^2}{n}\sigma_{\max}^2\big((I - S_\lambda)D^\dagger\big)$$
$$= \frac{(C_n')^2}{n}\max_{i=2,\ldots,n}\left(1 - \frac{1}{1 + \lambda\rho_i}\right)^2\frac{1}{\rho_i}$$
$$= \frac{(C_n')^2}{n}\lambda\max_{i=2,\ldots,n}\frac{\lambda\rho_i}{(1 + \lambda\rho_i)^2}$$
$$\leq \frac{(C_n')^2\lambda}{4n}.$$

In the third equality we have used the fact the eigenvectors of $I - S_\lambda$ are the left singular vectors of $D^\dagger$, and in the last inequailty we have used the simple upper bound $f(x) = x/(1 + x)^2 \leq 1/4$ for $x \geq 0$ (this function being maximized at $x = 1$).

Therefore, from what we have shown,

$$\sup_{\theta_0 : \|D\theta_0\|_2 \leq C_n'} \mathbb{E}\big[\mathrm{MSE}(\hat{\theta}^{\mathrm{LS}}, \theta_0)\big] \leq \frac{c_1\sigma^2}{n} + \frac{c_2\sigma^2}{\lambda^{d/2}} + \frac{(C_n')^2\lambda}{4n}.$$

Choosing $\lambda$ to balance the two terms on the right-hand side above gives $\lambda^* = c(n/(C_n')^2)^{2/(d+2)}$, for a constant $c > 0$. Plugging in this choice, and using upper bounds from the trivial cases $\lambda = 0$ and $\lambda = \infty$ when $C_n'$ is very small or very large, respectively, gives the result for Laplacian smoothing. $\square$

**Remark A.1.** *When $d = 4$, Lemma A.4 gives a slightly worse upper bound on $\sum_{i=1}^{n} 1/(1 + \lambda\rho_i)^2$, with an "extra" term $(nc_2/\lambda^{d/2}))\log(1 + c_3\lambda)$, for constants $c_2, c_3 > 0$. It is not hard to show, by tracing through the same arguments as given above that we can use this to establish an upper bound on the max risk of*

$$\sup_{\theta_0 \in \mathcal{S}_d(C_n')} \mathbb{E}\big[\mathrm{MSE}(\hat{\theta}^{\mathrm{LE}}, \theta_0)\big] \leq \frac{c}{n}\left((n\sigma^2)^{\frac{2}{d+2}}(C_n')^{\frac{2d}{d+2}}\log(n/(C_n')^2) \wedge n\sigma^2 \wedge n^{2/d}(C_n')^2\right) + \frac{c\sigma^2}{n},$$

*only slightly worse than the minimax optimal rate, by a log factor.*

*When $d \geq 5$, our analysis provides a much worse bound for the max risk of Laplacian smoothing, as the integral denoted $I(d)$ in the proof of Lemma A.4 grows very large when $d \geq 5$. We conjecture that this not due to slack in our proof technique, but rather, to the Laplacian smoothing estimator itself, since all inequalities the proof are fairly tight.*

## A.9 Utility lemmas used in the proofs of Theorems 5 and 6

This section contains some calculations on partial sums of eigenvalues of the Laplacian matrix $L$, for $d$-dimensions grids. These are useful for the proofs of both Theorem 5 and Theorem 6.

**Lemma A.3.** *Let* $L \in \mathbb{R}^{n \times n}$ *denote the graph Laplacian matrix of a $d$-dimensional grid graph, and* $\rho_{i_1,\ldots,i_d}$, $(i_1,\ldots,i_d) \in \{1,\ldots,\ell\}^d$ *be its eigenvalues, where* $\ell = n^{1/d}$. *Then there exists a constant* $c > 0$ *(dependent on $d$) such that, for any* $1 \le \tau \le \ell$,

$$\sum_{(i_1,\ldots,i_d) \in \{1,\ldots,\tau\}^d} \rho_{i_1,\cdots,i_d} \le c \frac{\tau^{d+2}}{\ell^2}.$$

*Proof.* The eigenvalues of $L$ can be written explicitly as

$$\rho_i = 4\sin^2\left(\frac{\pi(i_1-1)}{2\ell}\right) + \ldots + 4\sin^2\left(\frac{\pi(i_d-1)}{2\ell}\right), \quad (i_1,\ldots,i_d) \in \{1,\ldots,\ell\}^d. \quad \text{(A.8)}$$

This follows from known facts about the eigenvalues for the Laplacian matrix of a 1d grid, and the fact that the Laplacian matrix for higher-dimensional grids can be expressed in terms of a Kronecker sum of the Laplacian matrix of an appropriate 1d grid (e.g., [2, 6–9, 5]). We now use the fact that $\sin(x) \le x$ for all $x \ge 0$, which gives us the upper bound

$$
\begin{aligned}
\sum_{(i_1,\ldots,i_d) \in \{1,\ldots,\tau\}^d} \rho_{i_1,\cdots,i_d} &\le \frac{\pi^2}{\ell^2} \sum_{(i_1,\ldots,i_d) \in \{1,\ldots,\tau\}^d} \left((i_1-1)^2 + \ldots + (i_d-1)^2\right) \\
&\le \frac{\pi^2 d}{\ell^2} \tau^{d-1} \sum_{i=1}^{\tau} (i-1)^2 \\
&\le \frac{\pi^2 d}{\ell^2} \tau^{d-1} \tau^3 \\
&= \frac{\pi^2 d}{\ell^2} \tau^{d+2},
\end{aligned}
$$

as desired. $\square$

**Lemma A.4.** *Let* $L \in \mathbb{R}^{n \times n}$ *denote the graph Laplacian matrix of a $d$-dimensional grid graph, and* $\rho_i$, $i = 1,\ldots,n$ *be its eigenvalues. Let* $\lambda \ge 0$ *be arbitrary. For $d = 1, 2$, or 3, there are constants* $c_1, c_2 > 0$ *such that*

$$\sum_{i=1}^{n} \frac{1}{(1+\lambda\rho_i)^2} \le c_1 + c_2 \frac{n}{\lambda^{d/2}}.$$

*For $d = 4$, there are constants $c_1, c_2, c_3 > 0$ such that*

$$\sum_{i=1}^{n} \frac{1}{(1+\lambda\rho_i)^2} \le c_1 + c_2 \frac{n}{\lambda^{d/2}} \left(1 + \log(1 + c_3\lambda)\right).$$

*Proof.* We will use the explicit form of the eigenvalues as given in the proof of Lemma A.3. In the expressions below, we use $c > 0$ to denote a constant whose value may change from line to line. Using the inequality $\sin x \ge x/2$ for $x \in [0, \pi/2]$,

$$
\begin{aligned}
\sum_{i=1}^{n} \frac{1}{(1+\lambda\rho_i)^2} &\le \sum_{(i_1,\ldots,i_d) \in \{1,\ldots,\ell\}^d} \frac{1}{\left(1 + \lambda \frac{\pi^2}{4\ell^2} \sum_{j=1}^{d} (i_j-1)^2\right)^2} \\
&\le 1 + \int_{[0,\ell]^d} \frac{1}{\left(1 + \lambda \frac{\pi^2}{4} \sum_{j=1}^{d} x_j^2/\ell^2\right)^2} \, dx \\
&= 1 + c \int_0^{\ell\sqrt{d}} \frac{1}{\left(1 + \lambda \frac{\pi^2}{4} r^2/\ell^2\right)^2} r^{d-1} \, dr \\
&= 1 + c \frac{n}{\lambda^{d/2}} \underbrace{\int_0^{\frac{\pi}{2}\sqrt{\lambda d}} \frac{u^{d-1}}{(1+u^2)^2} \, du}_{I(d)}.
\end{aligned}
$$

In the second inequality, we used the fact that the right-endpoint Riemann sum is always an underestimate for the integral of a function that is monotone nonincreasing in each coordinate. In the third, we made a change to spherical coordinates, and suppressed all of the angular variables, as they contribute at most a constant factor. It remains to compute $I(d)$, which can be done by symbolic integration:

$$I(1) = \frac{\pi\sqrt{d}}{4\left(1 + \frac{\pi^2}{4}\lambda d\right)} + \frac{1}{2}\tan^{-1}\left(\frac{\pi}{2}\sqrt{\lambda d}\right) \leq \frac{1}{4} + \frac{\pi}{4},$$

$$I(2) = \frac{1}{2} - \frac{1}{2\left(1 + \frac{\pi^2}{4}\lambda d\right)} \leq \frac{1}{2},$$

$$I(3) = \frac{1}{2}\tan^{-1}\left(\frac{\pi}{2}\sqrt{\lambda d}\right) \leq \frac{\pi}{4}, \quad \text{and}$$

$$I(4) = \frac{1}{2}\log\left(1 + \frac{\pi^2}{4}\lambda d\right) + \frac{1}{2\left(1 + \frac{\pi^2}{4}\lambda d\right)} - \frac{1}{2} \leq \frac{1}{2}\log\left(1 + \frac{\pi^2}{4}\lambda d\right) + \frac{1}{2}.$$

$\square$

## Footnotes

[1]When $d = 1$, we have $m = n - 1$ edges, and so it is not be possible for $U$ to have orthonormal columns; however, we can just take its first column to be all 0s, and take the rest as the eigenbasis for $\mathbb{R}^{n-1}$, and all the arguments given here will go through.

[2]Here, albeit unconventional, it helps to index $\beta \in H(\tau) \subseteq \mathbb{R}^{n-1}$ according to components $i = 2, \ldots, n$, rather than $i = 1, \ldots, n-1$. This is so that we may keep the index variable $i$ to be in correspondence with positions on the grid.