[Reviews · NeurIPS 2016]

Reviewer 1

Summary

The paper deals with the estimation of functions defined over $n$ locations on a regular $d$-dimensional grid. The present work focuses on minimax rates on discrete TV class: Theorem 2 (and Theorem 3 respectively) states that under some condition on the radius $C_n$ of the TV class, the minimax rate is $C_n/n$ up to log factors (respectively $C_n^2/n$ if we restrict ourselves to linear estimators). The authors also generalize in Lemma 1 classical functional spaces embedding valid in dimension $1$ to dimensions $d \leq 2$ thanks to canonical choices of $C_n$ and $C_n'$. Thanks to this canonical choice of $C_n$, it turns out that TV denoiser are minimax optimal (up to log factor) over the TV class with radius $C_n$ for all $d \geq 2$ (see Eq (2)). This also generalizes the case $d=1$. A nice discussion on linear estimator is also provided, as well as their rates of convergence over Sobolev classes. The authors also provide minimax rates on discrete Sobolev class with radius $Cn'$.

Qualitative Assessment

The paper is very interesting. I learnt a lot reading it. I just report small typos in line 289.

Confidence in this Review

2-Confident (read it all; understood it all reasonably well)


Reviewer 2

Summary

This paper is concerned with the estimation of a function defined on a d-dimensional graph, when this function is assumed to belong to a TV or Sobolev class, and the risk is measured by the minimax normalized mean square error. After reviewing the known results for d=1, the authors obtain new results in larger dimensions: lower bounds for general and linear estimators, and upper bounds for fused lasso and trivial linear estimators.

Qualitative Assessment

The problem that is addressed here is interesting, and the authors provide a very significant amount of results. Now, I would like to say that I am not convinced by the claim that there is a canonical scaling Cn. This Cn was chosen because of technical considerations (relation between TV and Sobolev classes) and is not relevant in practice. It should be an unknown parameter of the model, and different upper bounds should be provided for every of the three lower bounds. On the other hand, the authors considered d and sigma as constants. I think that it could be more interesting to consider can be as large, respectively small, as possible. By the way, if d is fixed, I do not see the need for the notation dmax. On a related note, I think choosing rho1=1 in Theorem 2 works as well, as the constant c is arbitrary.

Confidence in this Review

2-Confident (read it all; understood it all reasonably well)


Reviewer 3

Summary

This paper establishes rates for TV denoising on grids, extending the work of Donoho & Johnstone and Hutter & Rigollet to dimension d > 1, with an emphasis on the case d=2.

Qualitative Assessment

- This paper is clearly a nice step in the understanding of TV denoising performance, but it lacks the optimality results in d >= 3 and/or on graphs different than a grid to be qualified as a major achievement. - The organization of the paper is well done, and it is easy to follow the results. Obviously, the submission guidelines impose to the authors to move the proof into a supplementary material, but the paper is readable nevertheless. - There is no mention of what happened when one consider a regression problem. I understand that it is way more challenging in this case, but can the authors identify where the methodology breaks ? - The proofs of the results are sound. Overall, I believe this paper is a good fit to NIPS, and should be accepted as an oral or poster.

Confidence in this Review

3-Expert (read the paper in detail, know the area, quite certain of my opinion)


Reviewer 4

Summary

The authors consider the problem of denoising a function f defined over a regular d-dimensional grid (such as a 2D image), where the available measurements are of the form y_i = f(i) + \epsilon_i \epsilon_i is i.i.d normal noise and the function f has bounded total variation (i.e. its sum of absolute differences along the edges is bounded) or belongs to a Sobolev class. The case of one-dimensional functions is already well-studied. The paper extends these results to more than one dimension, demonstrating qualitative differences between the d=1 case and d > 1. Their main results are: * Minimax lower bounds for denoising functions with bounded TV and functions from a Sobolev class. * Minimax lower bounds for linear denoisers acting on functions with bounded TV. * Proof that Laplacian eigenmaps and Laplacian linear smoothers obtain the minimax risk for Sobolev functions but not for functions with bounded variation. I think the most interesting result in the paper is that for functions in d > 1 dimensions with bounded variation, it is shown that TV denoising is minimax optimal (up to a log factor) whereas the best linear denoiser is not much better than some trivial estimators (identity and mean estimators). This is not the case for Sobolev functions, where simpler linear methods are proved to be minimax optimal.

Qualitative Assessment

This paper includes several results regarding the problem of function denoising over a grid of dimension > 1. In particular, demonstrating that simple linear methods are not sufficient for denoising functions of bounded variation and that TV denoisers are minimax optimal (up to a log factor). Overall, I feel these are solid contributions which should be of interest to the mathematical statistics, machine learning and signal processing communities. My main criticism is that the simulations shown in Figure 3 of the main text and Figure 1 of the supplementary use rather artificial signals. The paper would be much stronger if simulations on real data sets would be performed (such as 2D images, 3D volumetric scans, etc.) and see real-world performance of TV denoising vs. Laplacian smoothing and Laplacian eigenmaps. At its current form, the paper is rather theoretical and may be a better fit in a mathematical statistics journal or a theory conference such as COLT, both in contents and length. Minor issues and suggestions: * The 2016 manuscript by Hutter and Rigollet (reference [16]) has some closely related results. The authors should put more effort to contrast their results to those in that paper. * Table 1 is very nice. I think it would be great to see an expanded table that contains both the minimax rates for various dimensions and the rates obtained by TV denoising, Laplacian smoothing and Laplacian eigenmaps. * Proof of Lemma A.1 in the supplementary: can you explain the upper bound on ||D\theta||_1 in terms of the largest column vector of D? * Supplementary, line 34: "which completes the proof". This is a little confusing, since the \sigma^2 / n term (as stated in Theorem 3) is missing. * Proof of Lemma A.3: could you explain the last inequality? Typos: * Line 35: change "and in particurlarly optimality" to "and in particurlar optimality" * Line 77: change "where the infimum on in" to "where the infimum is on" * Line 171: change "contaning" to "containing" * Supplementary line 27: change "|D\theta||_1" to "||D\theta||_1"

Confidence in this Review

2-Confident (read it all; understood it all reasonably well)


Reviewer 5

Summary

This paper considers optimal minimax rates for denoising signals defined on graphs. They prove an unbridgeable gap between rates for linear regularizers like the mean, and the Laplacian smoothing, and total-variation regularization. This confirms intuitions about the superior properties of TV, with quantified bounds. The work extends works previously known in 1D, to higher dimensions. Among many possible applications, one can point-out brain-decoding via fMRI data (measured on a regular 3D grid).

Qualitative Assessment

The paper is very well-written. Tight links with existing literature are clearly exhibited. Rigorous proofs of the nontrivial claims are given, and complement a very extensive display of experimental results. The paper represents a tangible contribution to the difficult topic of signal denoising via penalized models.

Confidence in this Review

2-Confident (read it all; understood it all reasonably well)


Reviewer 6

Summary

The authors answer some open questions about the statistical performance of TV denoising in higher dimensions: They provide new lower bounds for classes of bounded total variation in higher dimensions, as well as lower and matching upper bounds for linear estimators over the same class. Their results show that the performance of linear estimators can be significantly worse than TV denoising if one picks a suitable n-dependent scaling for the total variation, which can be motivated by asking for a natural embedding of the TV class into other regularity classes to hold. Moreover, they give lower bounds for Sobolev classes in higher dimensions and show that contrary to the TV case, these are attained by linear estimators. Finally, they pose the question of whether TV denoising manages to achieve the minimax risk for Sobolev classes in dimension d >= 3 the same way it does for d <= 2.

Qualitative Assessment

Novelty The authors did a fine job of completing the picture of statistical minimax rates for total variation classes. Since TV denoising is heavily used in 2D, any results pertaining to that regime are important. Maybe the most interesting contribution is their establishing the “right” scaling (which is not constant) to be used for the variation term to allow comparison with other derivative based regularity classes. After this has been done, most of the results are interesting, but can be shown with standard techniques together with new estimates of spectral statistics of the grid graph. For example, the minimax rate for the TV class simply follows from the observation that the l_1 ball is embedded in it, so it can be immediately reduced to that case which is already available in the literature. I also agree with the authors on the conjectured minimax performance of TV denoising for Sobolev classes in higher dimensions being intriguing. Quality The proofs are sound and are executed with sufficient level of detail to be understood by someone with limited background in that field. It would have been even nicer to provide a bit of background on some of the concepts involved in the proofs, e.g. (linear) minimax rates for quadratically convex sets, but since this is restricted to the proofs in the supplementary material, this choice for the sake of brevity might be appropriate. The comments and discussion are helpful in order to appreciate the formulation of the results. Impact The paper offers some nice insight into theoretical performance bounds for a widely used denoising algorithm, which will be of interest to the imaging community as well as theoretical statistics. On the other hand, it does not offer new algorithmic insights nor fundamentally new proof techniques. However, it is notable that most of their lower bound results hold for arbitrary graphs. Also, the authors offer some venues for further exploration such as a possible generalization to trend filtering and investigating the performance of TV denoising over Sobolev classes in higher dimensions. Clarity The introduction and motivation sections are very well written and succeed in motivating the results that follow. The presentation of the results is clear and the comments helpful in putting them into context. Minor comments/corrections: Main paper: Line 4: parametrized _by_? Supplement: Line 8: “and in the second”: That seems to be dangling here. Display following line 39: “C_n” n missing Display following line 49, third to last line: Parentheses around the summands would be helpful Display following line 63, first line: Is it actually an equality here? Line 64: Donoho’s paper could already be referenced with a number here. Second display following 68: Variable clash for i_j, used twice in different contexts Display following line 82: The paper uses k instead of \tau. Also, it should be \theta instead of \beta. Third line of display following line 82: Here, something about the nullspace is being used to guarantee that we can write I \simeq D D^\dag. This should be explained in slightly more detail. Display following line 90: Again, the nullspace has to be well-behaved to be able to write this.

Confidence in this Review

2-Confident (read it all; understood it all reasonably well)